# Inflammation and the Potential Implication of Macrophage-Microglia Polarization in Human ASD: An Overview

**DOI:** 10.3390/ijms24032703

**Published:** 2023-01-31

**Authors:** Nadia Lampiasi, Rosa Bonaventura, Irene Deidda, Francesca Zito, Roberta Russo

**Affiliations:** Istituto per la Ricerca e l’Innovazione Biomedica IRIB, Consiglio Nazionale delle Ricerche, Via Ugo La Malfa 153, 90146 Palermo, Italy

**Keywords:** macrophage, mast cells, polarization, cytokines, exposome, MIA, microglia, innate immune system, ASD

## Abstract

Autism spectrum disorder (ASD) is a heterogeneous collection of neurodevelopmental disorders, difficult to diagnose and currently lacking treatment options. The possibility of finding reliable biomarkers useful for early identification would offer the opportunity to intervene with treatment strategies to improve the life quality of ASD patients. To date, there are many recognized risk factors for the development of ASD, both genetic and non-genetic. Although genetic and epigenetic factors may play a critical role, the extent of their contribution to ASD risk is still under study. On the other hand, non-genetic risk factors include pollution, nutrition, infection, psychological states, and lifestyle, all together known as the exposome, which impacts the mother’s and fetus’s life, especially during pregnancy. Pathogenic and non-pathogenic maternal immune activation (MIA) and autoimmune diseases can cause various alterations in the fetal environment, also contributing to the etiology of ASD in offspring. Activation of monocytes, macrophages, mast cells and microglia and high production of pro-inflammatory cytokines are indeed the cause of neuroinflammation, and the latter is involved in ASD’s onset and development. In this review, we focused on non-genetic risk factors, especially on the connection between inflammation, macrophage polarization and ASD syndrome, MIA, and the involvement of microglia.

## 1. Introduction

The first description of the pathology [1] has been overcome by the current Diagnostic and Statistical Manual of Mental Disorders (DSM-5) (APA, 2013), which describes the criteria for the so-called autism spectrum disorder (ASD). These include a heterogeneous collection of neurodevelopmental disorders characterized by early onset behavioral and social communication deficits associated with repetitive stereotypical behaviors. The genetic origins represent the main field of research considering the high estimates of heritability in ASD [2,3], as between 600 and 1200 genes associated with ASD have already been identified [4]. Additionally, many studies have highlighted the involvement of non-genetic factors in the physiopathology of ASD [5,6], such as air pollutants and pesticides [7,8,9], immune dysfunction [10] and other prenatal risk factors [3,11], as well as the gut microbiote [12] and microglia activation [13]. Many of these factors are among the categories forming the exposome, which consists of the multitude of environmental hazards to which people are exposed throughout their life, from conception to death. The exposome can lead to a wide range of human health dysfunctions, including neurodevelopmental problems such as in ASD. On the other hand, infections contracted in humans and rodents during pregnancy are known to result in maternal immune activation (MIA), which in turn can affect fetal neurodevelopment, resulting in an increased risk of developing central nervous system (CNS) disorders [3,11,14]. MIA induces activation of many cellular components of the innate and adaptive immune systems [15], and a maternal storm of pro-inflammatory cytokines and chemokines, which sustain inflammation [16,17]. Among the cells involved in MIA, macrophages/microglia exhibit plastic characteristics, undergoing the so-called polarization, i.e., changes in distinct functional phenotypes (M1/M2) in response to environmental stimuli. Macrophages/microglia’s polarization plays several diverse roles in tissue repair and homeostasis maintenance, as well as in numerous neurodegenerative diseases, in autoimmunity, atherosclerosis, and cancer; its dysregulation can cause chronic inflammation and pathological conditions [18,19].

To our knowledge, to date, there is only one study that directly correlates the polarization of maternal macrophages with ASD in humans [20], and very few investigating the fetal/childhood myeloid lineage activation/polarization in ASD (see “MIA fetal/childhood involvement” and “Microglia” paragraphs). Other data can be deduced from investigating maternal or fetal cytokines’ production (see “MIA maternal involvement” and “MIA fetal/childhood involvement” paragraphs).

The main topics addressed by this review are the role of the exposome, MIA and microglia in the onset of ASD. In particular, its purpose is to analyze the most recent studies concerning the implication of the myeloid lineage and their pro-inflammatory cytokines on the risk of developing ASD in humans, focusing on the potential macrophage switch between M1 and M2 status, where reported.

## 2. Exposome

### 2.1. Background

During their lifetime, people experience exposure to a multitude of environmental hazards found in water and air (i.e., pollution from pesticides, heavy metals, ultraviolet radiation, temperature) as well as in food and consumer products. The whole environmental exposure that people are subjected to from conception to death (all non-genetic factors, also including lifestyle factors) is called the “exposome”.

Wild [21] described three general categories for the exposome: (1) “processes internal to the body”, indicating the endogenous status of the organism (metabolism, gut microbiota, inflammation, lipid peroxidation, oxidative stress and aging, pregnancy complications); (2) “specific external exposures” (chemical contaminants and environmental pollutants, radiation, infectious agents, drugs, smoke); and (3) “general external exposures” (psychological, social, and economic influences on the individual, as lifestyle factors, medical interventions, physical activity, diet, climate, noise and light) (Figure 1). The three categories overlap and/or intertwine, and a complete exposome results from the combination of the internal environment of the body, the contexts (social, cultural, ecological) of people’s lives and the specific external agents of exposure. The highly dynamic nature of the exposome is among its challenging features, as it varies spatiotemporally and is diverse for each individual.

### 2.2. General External Exposome

The general external exposure considers several factors present in the environment in which we live, including climate, urban milieu, and lifestyle (physical activity, diet) as well as the socio-economic circumstances that physically and psychologically affect individuals. Despite numerous studies, it is still challenging to apply a distinction between exposure and response for social and psychological factors, and to find a direct cause–effect between them. Among the general external exposome factors, urbanization has undoubtedly brought several socio-economic benefits, but the urban milieu has also provided high levels of environmental hazards often associated with adverse health outcomes, especially in early life [22].

### 2.3. Specific External Exposome

Over the years, considerable interest has focused on the effects of human exposure to environmental pollutants, chemical contaminants, and radiation, all of which belong to the “specific external exposures” category. A critical and very sensitive phase of the human life cycle concerns pregnancy as, for example, it is increasingly recognized that maternal exposure to air pollution and environmental chemicals during pregnancy can affect both the health of the mother and the offspring, with possible outcomes even in the long-term, although the underlying mechanisms still remain mostly unknown [23].

Two different lines of research have shown one the link between air pollution exposure and a broad range of health dysfunctions [24,25], including also neurodevelopmental damages like ASD [26], the other the induction of an inflammatory response because of air pollution exposure [27] (Table 1). In particular, the possible role of exposure to air pollutants in the etiology of ASD had been suggested by numerous epidemiologic studies [28]. Within the framework of the Early Markers for Autism (EMA) study, Volk et al. [29] analyzed any potential relationship among the circulating levels of maternal cytokines/chemokines during pregnancy, the air pollution exposure in the month prior to the maternal serum collection, and their interplay in the risk of ASD in the future child. Although well-defined and clear relationships have not yet been established, this study is interesting as it began to address this critical issue by providing the necessary tools [29] (Table 1).

A recent study employed high-resolution metabolomics on neonatal blood spots to examine the metabolic effects of maternal exposure to traffic-related air pollution during pregnancy, known to raise the risk of adverse health outcomes, both pregnancy complications and birth and childhood disorders [30] (Table 1). Metabolomic changes were found, including alterations in the levels of pro- and anti-inflammatory metabolites mainly belonging to lipid and amino acid metabolisms, which were related to inflammation and oxidative stress pathways [23,31]. This study provided further evidence for the already known problem of the long-lasting negative effects of air pollution on developing organisms.

As a part of a European study (the HELIX project), researchers wanted to characterize the “early life exposome”, and one study reported the association of prenatal exposure to mercury (Hg) with pro-inflammatory cytokine responses (increased concentrations of interleukin (IL)-1β (IL-1β), IL-6, IL-8 and tumor necrosis factor-α (TNF-α)), a sign of ongoing inflammation, and an increased risk of liver damage in children [32]. Lead (Pb) intoxication is a potential etiological factor for the development of a chronic inflammatory response in ASD, as it induces mitochondrial dysfunction, reactive oxygen species (ROS) overproduction, and activation of a T-cell-dependent immune response with the production of pro-inflammatory cytokines (TNF-α) [33].

### 2.4. Internal Exposome

The state of health of the mother is an essential condition for the correct neurological development of the fetus. Important risk factors for the health of the fetus are linked to exposure, during pregnancy, to an unbalanced diet, a lack of some essential nutrients, the presence of hypertension, diseases such as diabetes, and of course infection/inflammation that will be discussed later (see “MIA” paragraph).

For example, many studies highlight the relationship between maternal exposure to nutrients (diet) and neurodevelopment outcomes. In fact, the diet followed by the mother during preconception, conception and pregnancy is fundamental for a correct development of the nervous and immunological systems of the fetus through epigenetic mechanisms [34,35,36,37].

In general, however, all those pathological events attributable to the onset of an acute or chronic inflammatory state in the mother constitute the real point of the risk of developing neurological disabilities [35]. In addition, various types of dysfunctions related to the internal exposome have been highlighted in ASD patients, such as dysregulation of the amino acid (AA) metabolism or alteration of the gut microbiota [36,37,38,39]. Indeed, AAs have key roles in numerous pathways related to the brain functions, such as metabolic intermediates or neurotransmitters. Moreover, there is evidence on the reciprocal interaction between the pediatric gut microbiome and the developing brain, namely the gut–brain axis, through various signaling molecules produced by both systems.

## 3. MIA

### 3.1. Background

Infection, inflammation and stress during pregnancy may activate the maternal immune system and can affect the development of the fetus [40,41]. MIA may result in activation of monocytes, macrophages, T cells and mast cells, involving the production of pro-inflammatory molecules, chemokines and cytokines. All these molecules can cross the placenta and the blood–brain barrier (BBB) with negative consequences for the neural development of the fetus [42,43]. Indeed, several studies have shown that pathogenic and non-pathogenic MIA (autoimmune and atopic diseases) may play a role in neurodevelopmental pathologies such as autism, schizophrenia and learning disabilities [44].

In general, the idea that infections, inflammation, and atopic disorders caused by the exposome (first hit) contribute to altering the maternal and fetal immune system and the fetal nervous system’s development has been consolidated. Moreover, the fetal nervous system becomes more susceptible to a subsequent secondary exposome hit, which could heavily impair its functionality [45,46,47] (Figure 2).

### 3.2. MIA Maternal Involvement

Maternal infection during pregnancy occurs in approximately 60% of women [48]. A large population-based study in the USA concerned the potential association of maternal infection during pregnancy accompanied by fever with ASD of the offspring [49,50]. The results showed that women who had an infection accompanied by fever during the second trimester of pregnancy were more likely to have children with ASD, compared to the general population [50] (Table 2). Fever is a physiological body response to infection and inflammation that permits the restoration of homeostatic conditions. During fever, many cellular components of the immune system, such as monocyte-macrophages [51] and mast cells [52], are activated, and both can produce pro-inflammatory and anti-inflammatory cytokines. In particular, IL-6 is associated with the induction of fever in the case of infection and inflammation [53], and with ASD [54].

Mac Giollabhui [55] investigated the relationship between the time of exposure to maternal infection (first trimester T1 and second trimester T2 were analyzed) with the risk of developing psychiatric diseases. The study was conducted on a cohort of 737 mothers and their offspring (aged 9 to 11 years) by measuring the maternal blood levels of some pro-inflammatory cytokines. The results demonstrated higher levels of IL-6, IL-8, tumor necrosis factor receptor-II (sTNF-RII), and IL-1 receptor antagonist (IL-1ra) in mothers of children showing psychiatric symptoms [55] (Table 2).

The cytokine IL-6 is the inflammation hallmark, produced mainly by M1 macrophages and activated mast cells; it is able to cross the placenta [56] and can influence fetal brain development in animal models [57]. Interestingly, IL-6 levels were measured in maternal serum in a cohort of 86 women during pregnancy (early, mild, and late), and this study correlated them with the structural and functional characteristics of offspring’s brain at birth. In particular, some characteristics of the amygdala, a region of the brain where emotional and stress behaviours are controlled, were taken into account [58]. The results suggested that maternal inflammation enhances the risk of developing psychiatric disorders, in part by altering the amygdala itself [58] (Table 2).

Non-pathogenic MIA may contribute to the development of abnormal behavior of the child as well. As an example, maternal atopic dermatitis (AD) during late pregnancy was associated with increased gestational IL-13 concentrations and polyunsaturated fatty acids (PUFAs)’ specific profiles. In turn, IL-13 and PUFAs levels were associated with increased risk of children’s behavioral difficulties (hyperactivity/inattention, emotional symptoms, conduct problems) [59]. IL-13 is a critical modulator in the CNS and cognitive function, and influences the development of the offspring’s brain by causing alterations in the neuron–microglia interaction [60] (Table 2). Patel showed that offspring of mothers with a history of non-pathogenic MIA (mainly asthma), displayed a higher severity of the behavioral disorders than the control group, while no differences were observed between the two groups regarding cognition disorders [61] (Table 2).

More recently, a large population-based study in Taiwan investigated the association between first-degree parental allergic and autoimmune diseases with the development of ASD and attention-deficit/hyperactivity disorder (ADHD) among children [62]. This study concerns three allergic disorders (e.g., asthma, allergic rhinitis, or atopic dermatitis) and four autoimmune diseases (e.g., rheumatoid arthritis, Sjogren’s syndrome, psoriasis, and systemic lupus erythematosus (SLE)). The results showed a significant association between allergic diseases of first-degree relatives and ASD of children, and among the autoimmune diseases, only the SLE of the siblings was significantly associated with the ASD of the children [62]. Mast cells and M1 macrophages are involved in allergic reactions, and asthma with the production of histamine, TNF-α and IL-6 [63,64,65], which can cross the placental barrier and can alter the BBB integrity [66].

In addition to maternal infections, life adversities occurring during pregnancy, such as maternal depressive symptoms, single-parent families, and lower socioeconomic status, showed significant association with children’s emotional health and ASD [67,68,69,70]. Different studies have highlighted a transcriptional profile indicative of increased immune activation, with upregulation of hallmark genes of macrophage polarization and slower fetal maturation (brain, heart, and immune system development) in low-income women [20,71,72] (Table 2).

All these are variables that can influence the mother’s physical and mental health and can act as multipliers of the risk of having children with psychiatric disorders.

MIA can induce the production of IL-17a by Th17 cells present in the placenta, and this cytokine may affect fetal development indirectly by regulating the placental function and the production of factors that can cross the placenta [73,74]. Casey and colleagues analyzed maternal serum levels of IL-17a and found significantly reduced expression at mid-gestation (20 weeks) in mothers of children with ASD [75] (Table 2).

### 3.3. MIA Fetal/Childhood Involvement

Alterations in innate and adaptive immune cells [76] as well as increased levels of several pro-inflammatory cytokines, including IL-1β, IL-6, TNF-α, monocyte chemoattractant protein-1 (MCP-1), and IL-8, have been shown in the brain and cerebrospinal fluid of many ASD patients [77] (Table 3). Furthermore, high plasma levels of IL-1β, IL-6, IL-8, and IL-17 [77,78], homocysteine (HCY) and C-reactive protein (CRP) [79], and dysregulation of the antioxidant system [80] were found in children showing ASD and behavioral alterations.

A recent study analyzed the activation of circulating CD14+ monocytes triggered by toll like receptor (TLR)2 and TLR4 stimulation in ASD patients [81] (Table 3). Higher levels of IL-6 production, after TLR4 stimulation, were seen in ASD monocytes with respect to healthy children, and the higher levels also correlated with worsening behaviors. This dysfunctional activation of myeloid cells, which can include monocyte-macrophages and microglia, may indicate dysregulation of immune response after activation. On the contrary, the correlation of higher levels of IL-6 with worse behaviors indicates the relationship between pro-inflammatory cytokines and ASD [81]. Mac Giollabhui [82,83] reported that higher levels of IL-6, TNF-α and CRP were associated with psychiatric disorders in adolescent and adult subjects, and determined a link to low socioeconomic status, thus suggesting that demographic variables play an important role (Table 3). IL-6, mainly produced by monocytes, macrophages M1 and activated mast cells, leads to an inflammatory activation cascade that drives downstream innate and adaptive immune responses, including T cell activation and expansion. The possible mechanisms affected in the brain by IL-6 dysregulation are impaired cell adhesion and migration, and consequently, improper formation of synapses, imbalances in excitation (glutamate-mediated) and inhibitory (gamma-aminobutyric acid (GABA)-mediated) neurotransmission [84]. No studies before Yamauchi had focused on polarized macrophages (M1 and M2) of patients and their role in ASD disease [85] (Table 3). These authors induced in vitro the M1 and M2 polarization of blood peripheral monocytes isolated from ASD patients into macrophages, and then measured the expression levels of both pro-inflammatory and anti-inflammatory cytokines. The only significantly expressed cytokine was the TNF-α, which was markedly higher in M1 macrophages from ASD patients with respect to control subjects. Interestingly, resting monocytes did not show any differences between ASD children and control subjects [67]. TNF-α produced by M1 macrophages could have similar effects in the CNS, as TNF-α produced by microglia (i.e., macrophages of the CNS) has been reported to impair synaptogenesis [86]. Furthermore, recent findings indicated that perivascular macrophages, regularly renewed by monocytes, are critical for controlling brain functions [87,88]. Thus, like TNF-α produced by microglia, macrophage-derived TNF-α likely weakens brain function and may play a role in the development of neuropsychiatric disorders. Mast cells reside in the brain and are involved in neuroinflammation responding to neuropeptides (neurotensin (NT) and substance P (SP)), and in turn produce pro-inflammatory cytokines IL-6 and TNF-α through the nuclear factor-κB (NF-κB) pathway [66,89,90]. In addition, activated mast cells can damage the BBB, allowing the entry of large molecules (TNF-α) and cells (monocyte-macrophage), which enhances neuroinflammation [66]. Higher levels of TNF-α, IL-17a, and IL-23 increased the permeability of the BBB in mice [91]. In addition, IL-17a stimulates neuroinflammation through its IL-17a receptor (IL-17ra), and the IL-17ra blockade may reduce monocyte-associated oxidative stress [92]. ASD patients showed increased IL-17 and IL-17ra expression in their monocytes associated with increased NF-κB expression, and activity and inducible nitric oxide synthase (iNOS) expression compared to typically developing children [92] (Table 3).

In a systematic study conducted on the Gene Expression Omnibus (GEO) database, ASD children showed higher levels of infiltrate immune cells such as plasma cells, monocytes, neutrophils, naïve CD4 T cells, and activated mast cells, dendritic cells and CD4 memory T cells compared to control subjects [93]. In addition, the authors identified molecular subtypes based on ferroptosis-related genes (FRGs)’ expression [93]. Ferroptosis is a recently reported novel non-apoptotic Fe2+-dependent cell death characterized by increased intracellular oxidative stress NF-E2–related factor 2 (Nrf2)-dependent, which induces lipid peroxide oxidation, and glutathione (GSH) depletion, accompanied by inflammatory reactions and macrophages’ polarization [94] (Table 3). As previously reported, alteration of antioxidant networks has been found linked to dysregulation of monocytes in ASD patients, suggesting that oxidative stress plays a role in the pathogenesis of ASD [95]. The IL-1β/IL-10 ratio has been used to subgroup patients with ASD who have different symptoms (normal, low, high), since the ratio reflects the imbalance of pro-inflammatory and anti-inflammatory cytokines produced by monocytes, and is associated with the severity of the disease [96]. Macrophage migration inhibitory factor (MIF), a mediator of the immune response, can be produced from the monocyte-macrophage lineage [97] and has been proposed to play a role in the pathogenesis of ASD, regulating the expression of innate cytokines such as TNF-α [98,99] (Table 3). Recently, Ning et al. found that serum levels of MIF in the children with ASD are significantly higher than in control subjects, and could be used as a biomarker for the disease in association with IL-6 levels [100]. Analyzing ASD monocytes, Kutuk et al. found that the increased expression levels of IL-6, IL-17 and IL-1α covered 56.6% of ASD children, suggesting that these cytokines may be used as early markers of the pathology [101] (Table 3). A meta-analysis study, conducted in 2021 (including studies dated from 2002–2019), found that the levels of 13 different blood cytokines (i.e., IL-6, IL-1β, IL-12p70, MIF, eotaxin-1, MCP-1, IL-8, IL-7, IL-2, IL-12, TNF-α, IL-17, IL-4) were significantly altered in ASD patients with respect to the control group [102] (Table 3). In general, different studies have shown that the imbalance between pro- and anti-inflammatory cytokines and chemokines contributes to the dysregulation of immune homeostasis in patients with ASD [103,104] (Table 3). However, the specific mechanism of action of most of the altered cytokines in ASD is still unclear, and for this reason, further studies are needed; that said, there is no doubt that an immune dysregulation occurs.

In an attempt to find predictive and early ASD biomarkers, a proteomics study was recently conducted [105]. A group of 9 proteins were found to be significantly altered in the ASD subjects compared to the control group, using three different computational methods. These proteins contribute to some pathways that participate in the regulation of the response of the innate immune system [105] (Table 3). A very large number of proteins were examined, and the only limitation of this study is that the investigation did not consider gender-specific differences.

Individuals with ASD are very fragile subjects; they cannot tolerate any action that invades their body or their emotions. Hence, there is the need to find biomarkers useful for the early diagnosis of ASD that can be detected in a non-invasive way. This prompted researchers to look for new fluids, other than blood and urine, in which to measure the altered levels of the cytokines. A recent study investigated the cytokine content in the saliva of ASD patients showing that a panel of pro-inflammatory cytokines was increased compared to normal subjects [106] (Table 3).

In conclusion, MIA may induce autism-like behaviors in offspring, but the mechanism of action on the fetal brain is not yet clear. Growing evidence in animal models indicates an action on microglia and their activation/inflammatory polarization.

## 4. Microglia

### 4.1. Biological Function

Microglia are tissue-resident macrophages of the CNS that regulate brain homeostasis, and the first glial cells involved in host defense/infection of CNS through the process of antigen presentation, the phagocytosis of toxic products and the release of numerous inflammatory mediators. In addition, microglia have a broad array of physiological functions in the healthy brain, including the regulation of neurogenesis, myelination, and synaptic remodeling [107,108].

Under physiological conditions, the microglial morphology in a “resting” or inactivated state is typically ramified, and the number and the distribution of the microglial population are strongly regulated and are in a continuous state of surveillance by the surrounding microenvironment, carrying out neuroprotective effects [109]. In response to the challenges of the CNS microenvironment, the microglia undergo activation and proliferation, changing their morphology in ameboid with an enlarged cell body and retracted, thick processes, and assume the role of removing cellular debris, such as macrophages in the periphery, by phagocytosis [110]. Similarly to macrophage phenotypic changes, the reactive microglia can be differentiated into two polarization states: M1 and M2 phenotypes [111]. Generally, M1 microglia predominate at the injury site at the end stage of disease, causing a pro-inflammatory response with production of cytokines such as TNF-α and IL-1β, superoxide, nitric oxide (NO) and ROS, which can lead to neuroinflammation. On the contrary, the M2 microglia, activated in the presence of IL-4 or IL-13, are involved in the anti-inflammatory response directed towards immune resolution and tissue repair [112]. Their phagocytic capacity is accompanied by the production of anti-inflammatory cytokines such as IL-10 and transforming growth factor-β (TGF-β) [112]. Microglial dysfunction has been closely associated with several disorders, ranging from neurodegenerative diseases, such as Alzheimer’s disease (AD), Parkinson’s disease (PD) and amyotrophe lateral sklerose (ALS), to neurodevelopmental disorders, including schizophrenia (SCZ) and ASD [113,114,115,116,117].

### 4.2. Neuroinflammation

Neuroinflammation is defined as an inflammatory response within the brain or spinal cord. This inflammation depends on the activation of resident CNS glial cells (microglia and astroglia), endothelial and peripheral immune cells, and the production of their proinflammatory molecules. Neuroinflammation is an important contributor to ASD pathogenesis and in this context, the microglial population seems to have a key role (Figure 3).

In the last decades, the scientific interest has been directed to study the role of microglia and their dynamic changes of M1/M2 phenotypes in several neurodegenerative diseases such as AD, ALS and PD, whereas little on the role of microglia in ASD disease has been investigated, and only very few studies have been focused on the study of microglia polarization/activation in human ASD.

The activation of microglia can be influenced in both the prenatal and postnatal period, suggesting the two-hit theory [14]. In fetal life, MIA may increase the sensitivity of microglia to further events in postnatal life, leading to the development of the disease [14]. Prenatal infection and psychological trauma in postnatal life can act synergistically to increase the risk of developing SCZ disease [118].

MIA-induced neuroinflammation could affect fetal microglia, inducing changes in their phenotype and consequently inducing functional alterations (microglial, synaptic, and neurobehavioral dysfunctions) in adulthood [119,120,121]. However, the underlying mechanisms are largely unknown. Among the possible mechanisms leading to neuroinflammation in ASD children, it has been suggested that TNF-α might mediate inflammation in the brain by stimulating microglia to recruit monocytes, although the development of inflammation directly in the brain through glial cells cannot be excluded [33] (Table 4).

In humans, neuropathological studies conducted in ASD post-mortem tissues have revealed important characteristics of the microglial population. The first evidence of microglial activation in several brain tissues of children and adult ASD patients has been reported by Vargas [122]. A marked microglial involvement has been revealed in the cerebellum, and a significant increase in the expression levels of two pro-inflammatory chemokines (MCP–1 and thymus and activation-regulated chemokine (TARC)) and an anti-inflammatory cytokine (TGF-β) has been found in the brain of ASD patients [122] (Table 4). Other studies have demonstrated an imbalance between pro- and anti-inflammatory cytokines (TNF-α, IL-6, granulocyte-macrophage colony-stimulating factor (GM-CSF), interferon γ (IFN-γ), IL-8) [123] and a preponderant production of a pro-inflammatory cytokine (IL-6) in ASD and SCZ subjects [124,125]. Unfortunately, although they show evidence of the neuroinflammatory process in progress in the brain tissue of ASD patients, these last studies do not provide any information on the glial subtype involved (microglia and astroglia).

Microglia express different surface proteins that can be used as markers. These include ionized calcium-binding adapter molecule 1 (Iba-1), 18 kDa translocator protein (TPSO), CD11b and CD68. Iba-1 is a member of the calcium-binding protein group and is the most used marker for microglia. Further studies have observed an increase in the microglial density (studying the expression of Iba1) and a morphological change in the active form in multiple regions of cortex, including the prefrontal cortex (PFC), and the visual and front insular cortex from children and adults with ASD [126,127] (Table 4). The identification of microglia has been conducted in specific layers of PFC in ASD subjects, where changes in the number of neurons and astroglia have been reported compared to control subjects [128] (Table 4).

However, contradictory results are reported on microglia activation observed by studying Iba-1, TSPO and their localization inside the brain of ASD patients [129,130,131] (Table 4). These contradictory results depend on the non-exclusive expression of these molecular markers in microglia, as they are also expressed in monocytes and macrophages.

A transcriptomic study of post-mortem brains has shown a specifically enriched M2 microglial “signature” and “immune response” genes in ASD brains [132] (Table 4). In addition, the M2 microglial signature has been negatively correlated with a differentially expressed neuronal signature, suggesting its alteration in innate immunity and neuronal activity in the ASD disease [132]. Accordingly, a transcriptomic study conducted on post-mortem brains of neuropsychiatric and neurodegenerative patients (n = 2633) showed a dysregulation of many genes related to microglia, such as myelin protein zero like 2 (MPZL2), SERPINA, heat shock protein family A (Hsp70) member 6 (HSPA6) and gamma-aminobutyric acid type A receptor subunit epsilon (GABRE), in eight different brain regions [133] (Table 4).

Recently, single-nucleus RNA sequencing of cortical tissue from ASD patients has highlighted enriched expression of genes associated with microglia and astrocytes’ activation, as well as transcription factors regulating developmental processes and upregulation of cell motility (glycogen synthase kinase3 (GSK3), spleen tyrosine kinase (SYK), fyn binding protein (FYB)) [134] (Table 4). A new approach to studying data, based on systems biology and focused on molecular targets, could be useful for integrating data from patients and from experimental studies on animals. A clinical database of subjects with ASD was interrogated to investigate the presence of gene copy number variation (CNV) [135]. In the CNV gene frame (n = 659), 121 genes were highly expressed in the prenatal period in the brain, and concerned the signaling pathways involving the vascular system, neuroinflammation, and activation of microglia (Table 4). In particular, defects in genes that regulate the neurovascular system appear to correlate with symptoms of ASD. For this purpose, the authors used a knockout mouse model of the semaphorin 3F- neuropilin 2 protein (Sema 3F-NRP2 KO), involved as a neuronal guidance molecule in the developing brain in a wide variety of tissues including the immune and vascular systems, and demonstrated that in the brains of the mice presenting ASD symptoms, there was neuroinflammation, microglial activation, induction of iNOS and increased 3-nitrotyrosine, BBB deficiency, and disruption of neurovascular signaling [135].

Considering the studies conducted so far, activated microglia capable of producing pro-inflammatory cytokines play an important role in the development of neuroinflammation and disease, although many aspects are still to be clarified. The main difficulty is that studies can be conducted in the human brain only after death. On the other hand, animal studies, easier to perform, have highlighted the relationship between MIA, polarized microglia and the disease [136]. However, even if it is true that the conclusions obtained from the studies conducted on animals cannot be directly transferred into strategies/therapies for humans, they constitute an important starting point.

## 5. Conclusions

Our final considerations mainly concern the limitations of these studies. Many studies have a limited number of subjects with ASD pathology enrolled. Another weakness is the great difficulty of considering all the socio-demographic variables (race, ethnicity, age, economic level, etc.), the comorbidities present (epilepsy, anxiety, etc.), and the influence exerted by diet, pollution, and epigenetic and genetic factors, etc. In addition, other factors must be taken into consideration, such as the type of maternal immune dysregulation (infection bacterial/viral, autoimmunity, allergy, asthma), the severity (acute versus chronic infection/inflammation), the timing (early versus late pregnancy), and the duration of the exposure. For all these reasons, it is difficult to translate preclinical conclusions obtained from animals into human strategy.

Therefore, further studies are needed to understand the exact mechanisms and the reciprocal interactions of all the factors that may have a role in the development of ASD and cognitive alteration.

## Figures and Tables

**Figure 1 ijms-24-02703-f001:**
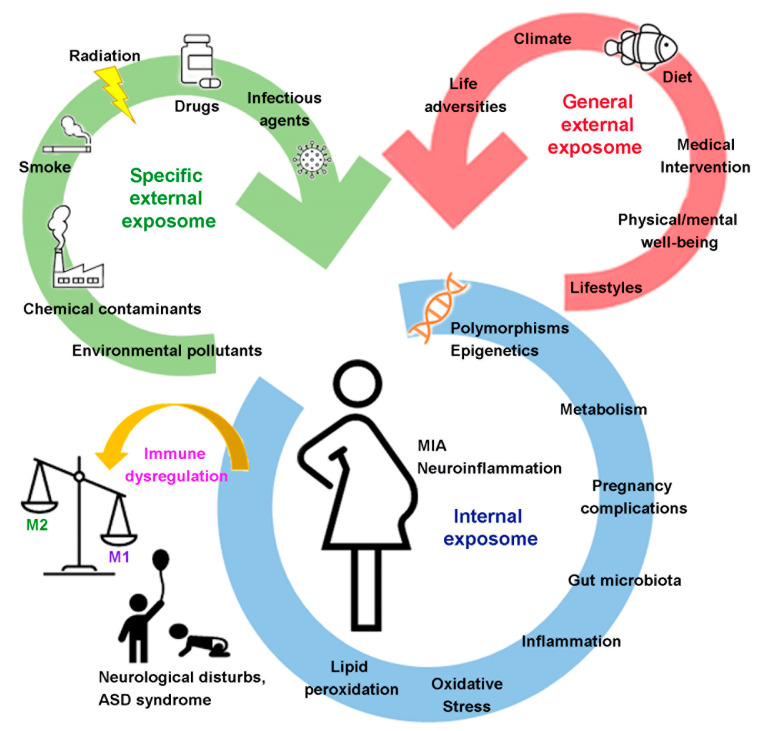
Schematic representation of the effects of different exposome factors on MIA, neuroinflammation and immune dysregulation occurring in the period of neurodevelopment.

**Figure 2 ijms-24-02703-f002:**
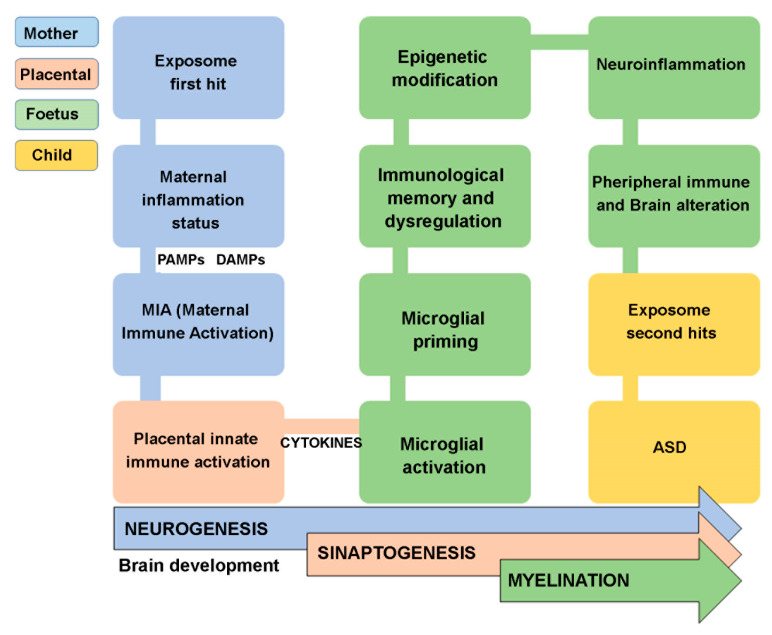
Schematic representation of the main steps occurring to develop ASD.

**Figure 3 ijms-24-02703-f003:**
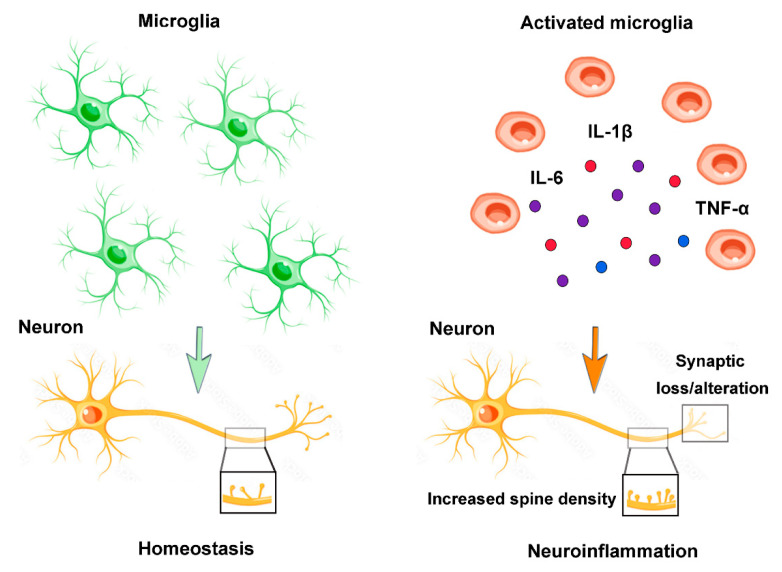
Microglia and neuroinflammation. During homeostasis, branching microglia play crucial roles such as surveillance, phagocytosis of debris, synaptic homeostasis, maintenance of neuronal plasticity, and trophic support. Neuroinflammation causes the microglial phenotype to change to an amoeboid state and the release pro-inflammatory cytokines (IL-6, TNFα, IL-1β), with loss of homeostatic functions, synaptic alterations, increased spinal density and neuronal dysfunction.

**Table 1 ijms-24-02703-t001:** Relationship between exposome and maternal/child pathological outcome.

Type of Exposure	Outcome	References
Air pollution	Neurodevelopment damage	[26,28]
Air pollution	Pro-inflammatory cytokines	[27,29]
Traffic-related air pollution	Pro-inflammatory metabolites	[30]
Chemicals	Pro-inflammatory metabolites	[23,31]
Hg, Pb	Pro-inflammatory cytokines	[32,33]

**Table 2 ijms-24-02703-t002:** Schematic list of maternal cytokines produced, stimuli received, and pregnancy time analyzed in studies related to ASD mentioned in this review. T1 (0–3 months), T2 (3–6 months), T3 (6–9 months).

Cytokine/System	Stimulus	Pregnancy Time	References
None	Infection + fever	T2	[50]
IL-6, IL-8, IL-1ra, IL-1β, sTNF-RII	Infection	T1 and T2	[55]
IL-6 and amygdala	None	T1, T2, T3	[58]
IL-13, PUFAs	Atopic dermatitis	T3	[59]
Pro-inflammatory cytokines	Asthma	after birth	[61]
Pro-inflammatory cytokines	Allergic and autoimmune diseases	after birth	[62]
Pro-inflammatory cytokines	Prenatal adversity	after birth	[20]
Pro-inflammatory cytokines	Low income	at term	[71]
Pro-inflammatory cytokines, NF-κb, AP1	Stress, low income	T3	[72]
IL-17a	Infection	T2	[75]

**Table 3 ijms-24-02703-t003:** Schematic list of the cytokines produced, and related tissues found in patients with ASD during the fetal/childhood period mentioned in this review.

Cytokine/System	Tissue/cell	References
IL-1β, IL-6, TNF-α, MCP-1, IL-8	Brain, cerebrospinal fluid	[76,77]
IL-1β, IL-6, IL-8, IL-17	Plasma	[77,78]
HCY, CRP	Plasma	[79]
Antioxidant system alteration	Monocytes	[80]
Il-6	Monocytes CD14+	[81]
IL-6, TNF-α, CRP	Plasma	[82,83]
TNF-α	Macrophages M1 and M2	[85]
IL-17, IL-17ra, NF-κB, iNOS	Monocytes	[92]
Antioxidant system alteration	Plasma cells, monocytes, neutrophils, naïve CD4 T cells, and activated mast cells, dendritic cells and CD4 memory T cells	[93]
IL-1β/IL-10 ratio	Monocytes	[96]
MIF	Serum	[100]
IL-6, IL-17,IL-1α	Monocytes	[101]
IL-6, IL-1β, IL-12p70, MIF, eotaxin-1, MCP-1, IL-8, IL-7, IL-2, IL-12, TNF-α, IL-17, IL-4	Blood	[102]
MIP-1a, MIP-1b	Blood	[103]
Pro-inflammatory/anti-inflammatory cytokines ratio	B cells	[104]
Protein of immune system regulation	Plasma/serum	[105]
Pro-inflammatory cytokines	Saliva	[106]

**Table 4 ijms-24-02703-t004:** Schematic list of the cytokines and microglia markers mentioned in this review, and potential roles found in ASD patients.

Cytokine/Marker	Tissue/Cell	References
TNF-α	Microglia activation and monocytes’ recruitment in human brain	[33]
MCP-1, TARC, TGF-β	Microglia activation in human	[122]
TNF-α, IL-6, GM-CSF, IFN-γ, IL-8	Up-regulation in human brain cortex	[123]
IL-6	Glial activation in human cerebellum	[124]
IL-6	Upregulation in human brain	[125]
Iba-1	Increased microglia density in human	[126,127]
None	Increased number of neurons and decreased of astroglia in human	[128]
Iba-1	Increased number of microglia in human TC and no variation in astroglia	[129]
TSPO	No change in human brain	[130,131]
M2 genes	Microglia activation in human	[132]
MPZL2, SERPINA, HSPA6, GABRE	Microglia activation in human	[133]
GSK3, SYK, FYB	Microglia, astroglia activation in human	[134]
121 genes	Microglia activation, dysregulation of vascular system, neuroinflammation	[135]

## Data Availability

Not applicable.

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
