# Peer review of "Inflammation and the Potential Implication of Macrophage-Microglia Polarization in Human ASD: An Overview"

_ijms, 2023, doi:10.3390/ijms24032703_

Round 1

Reviewer 1 Report

I have gone through the article submitted to journal for consideration. I have following suggestions for them:

1. The introduction should be shorten and clear. They have used very long sentences throughout the article. I recommend them to use shorter and precise sentences.

2. The article is not written according to its title. As lots of extra and irrelevant information has decreased its quality. So I will highly recommend authors to shorten the article to its half to focus their title. 

Author Response

We thank the reviewers for the helpful suggestions that improve our manuscript.

Reviewer 1

I have gone through the article submitted to journal for consideration. I have following suggestions for them:

  1. The introduction should be shorten and clear. They have used very long sentences throughout the article. I recommend them to use shorter and precise sentences.
  • We shortened and clarified the introduction and all the manuscript.
  1. The article is not written according to its title. As lots of extra and irrelevant information has decreased its quality. So I will highly recommend authors to shorten the article to its half to focus their title. 
  • We revised all the manuscript in accordance with the suggestions received. A lot of unnecessary information has been deleted; some parts have been moved for better reading.

Reviewer 2 Report

In the paper” Inflammation and the potential implication of macrophage-microglia polarization in human ASD: an overview”, the authors mentioned the impact of genetic and non-genetic influences on developing Autism spectrum Disorders (ASD), especially during pregnancy for the mother and her fetus. They also pointed out that activation of monocytes, macrophages, mast cells, and microglia and high production of pro-inflammatory cytokines may cause neuroinflammation, and the latter on it contributes to ASD onset and development. In addition to the fate of macrophages, microglia, etc., they narrated the possibilities of the pathogenic and non-pathogenic maternal immune activation (MIA) and autoimmune diseases to alterations in the fetal environment, and thus the etiology of ASD. The review is well-written and covers all the pieces of literature that fall into the field. 

Author Response

We thank the reviewers for the helpful suggestions that improve our manuscript.

Reviewer 2

 The manuscript “Inflammation and the potential implication of macrophage-microglia polarization in human ASD: an overview.” presented by N. Lampiasi et al. is interesting and comprehensive.

This author, N. Lampiasi, has previously published an Editorial in IJMS as a summary of a Special Issue titled “Macrophage Polarization: Learning to Manage It. Int. J. Mol. Sci. 2022, 23, 2–5, 976 doi:10.3390/ijms23137208”. This time present a review about macrophage-microglia polarization in human Autism Spectrum Disorder (ASD).

 I only have minor corrections/comments:

1) I expected a specific subsection for “specific external exposures” and “general external exposures” instead of a unique “2.2 External exposome section” where general external exposures are practically not commented on or reviewed.

  • As suggested by the reviewer we create subsections for “specific external exposures” and “general external exposures” and commented them.

2) I suggest indicating the numbers of the references in all Tables, instead of or additionally the name of the first author. It would be easier for readers to locate the references that are mentioned throughout the text. The way it is now, you have to go first to the bibliography section and then go back to the table.

  • We indicated the numbers of the references in the Tables.

3) Table 1. Define the acronym “EOAs” . It is defined (line 150) but after reference to Table 1.

  • According to the reviewer 1 we shortened the review and the sentences about EOAs was removed from the text and the table.

4) “IL-1ra” is also indicated as “IL-1rA” and “IL-17rA”. To avoid possible confusion, spell only in one way “IL-1ra” or “IL-1RA” (follow the same form for IL-17rA)

  • We indicated IL-1ra and IL-17ra in the text.

5) I, personally, consider it not necessary to indicate the tools used for figure generation (Power-point, …) but it is just my personal opinion.

  • We removed the indication of the tools used for figure generation.

Reviewer 3 Report

Manuscript report:    IJMS-2081948

The manuscript “Inflammation and the potential implication of macrophage-microglia polarization in human ASD: an overview.” presented by N. Lampiasi et al. is interesting and comprehensive.

This author, N. Lampiasi, has previously published an Editorial in IJMS as a summary of a Special Issue titled “Macrophage Polarization: Learning to Manage It. Int. J. Mol. Sci. 2022, 23, 2–5, 976 doi:10.3390/ijms23137208”. This time present a review about macrophage-microglia polarization in human Autism Spectrum Disorder (ASD)

 I only have minor corrections/comments:

 1) I expected a specific subsection for “specific external exposures” and “general external exposures” instead of a unique “2.2 External exposome section” where general external exposures are practically not commented on or reviewed.

 2) I suggest indicating the numbers of the references in all Tables, instead of or additionally the name of the first author. It would be easier for readers to locate the references that are mentioned throughout the text. The way it is now, you have to go first to the bibliography section and then go back to the table.

 3) Table 1. Define the acronym “EOAs” . It is defined (line 150) but after reference to Table 1.

 4) “IL-1ra” is also indicated as “IL-1rA” and “IL-17rA”. To avoid possible confusion, spell only in one way “IL-1ra” or “IL-1RA” (follow the same form for IL-17rA)

 5) I, personally, consider it not necessary to indicate the tools used for figure generation (Power-point, …) but it is just my personal opinion.

Author Response

We thank the reviewers for the helpful suggestions that improve our manuscript.

 Reviewer 3

In the paper” Inflammation and the potential implication of macrophage-microglia polarization in human ASD: an overview”, the authors mentioned the impact of genetic and non-genetic influences on developing Autism spectrum Disorders (ASD), especially during pregnancy for the mother and her fetus. They also pointed out that activation of monocytes, macrophages, mast cells, and microglia and high production of pro-inflammatory cytokines may cause neuroinflammation, and the latter on it contributes to ASD onset and development. In addition to the fate of macrophages, microglia, etc., they narrated the possibilities of the pathogenic and non-pathogenic maternal immune activation (MIA) and autoimmune diseases to alterations in the fetal environment, and thus the etiology of ASD. The review is well-written and covers all the pieces of literature that fall into the field. 

  • We thank the reviewer for his/her comments.

Round 2

Reviewer 1 Report

Authors have improved the article. However, I further suggest them following updates:

1. Table 1 should be shortened by combining exposures of same types, like air pollution (indoor and outdoor), chemicals combine mercury and lead (mention complete name with chemical symbols). 

2. On page 4 line 112 ASD is mentioned as ADS, kindly correct it.

3. MIA abbreviation should be explained when used for the first time in the text.

4. Maternal infection during pregnancy occurs in approximately 60% of women. Provide reference for the statement.

5. In table 4 it should Iba-1correct it and also explain the abbreviation.

6. I will recommend authors to explain all abbreviation sued first time in the text at place. Give specific attention to abbreviations used in table 4.

Author Response

Authors have improved the article. However, I further suggest them following updates:

1. Table 1 should be shortened by combining exposures of same types, like air pollution (indoor and outdoor), chemicals combine mercury and lead (mention complete name with chemical symbols).

 A. In accordance with the reviewer's suggestion in Table 1 we combine mercury and lead which evoke the same result (outcome). Concerning Air Pollution, Traffic Air Pollution and Chemical Derived Pollution as they do not cause the same result (outcome), we would like to highlight them as separate events. We mentioned complete name and chemical symbols for mercury and lead.

Air pollution

Neurodevelopment damage

[26, 28]

Air pollution

Pro-inflammatory cytokines

[27, 29]

Traffic-related air pollution

Pro-inflammatory metabolites

[30]

Chemicals

Pro-inflammatory metabolites

[23, 31]

2. On page 4 line 112 ASD is mentioned as ADS, kindly correct it.

A. We corrected it.

3. MIA abbreviation should be explained when used for the first time in the text.

A. We have removed extra MIA abbreviation.

4. Maternal infection during pregnancy occurs in approximately 60% of women. Provide reference for the statement.

A. We added reference (48).

5. In table 4 it should Iba-1correct it and also explain the abbreviation.

A. We corrected it.

6. I will recommend authors to explain all abbreviation sued first time in the text at place. Give specific attention to abbreviations used in table 4.

A. We have gone through the whole manuscript and added all the necessary abbreviations.
